# Antioxidant Activity of Selected Medicinal Plants Used by Traditional Herbal Practitioners to Treat Cancer in Malawi

**Friday Fosta Fred Masumbu** [1], **John Finias Kamanula** [1], **Anthony Mwakikunga** [2], **Bonface Mwamatope** [3] **and David Tembo** [4,*]

[1] Chemistry Department, Faculty of Science, Technology and Innovation, Mzuzu University, P/B 201, Luwinga, Mzuzu 2, Malawi; masumbu.f@mzuni.ac.mw (F.F.F.M.); kamanula.j@mzuni.ac.mw (J.F.K.)

[2] Biomedical Sciences Department, School of Life Sciences and Allied Health Professions, Kamuzu University of Health Sciences, P/B 360, Chichiri, Blantyre 3, Malawi; amwakikunga@kuhes.ac.mw

[3] Basic Sciences Department, Faculty of Agriculture, Lilongwe University of Agriculture and Natural Resources, Lilongwe P.O. Box 219, Malawi; bmwamatope@luanar.ac.mw

[4] Physics and Biochemical Sciences Department, Faculty of Applied Sciences, Malawi University of Business and Applied Sciences, P/B 303, Chichiri, Blantyre 3, Malawi

[*] Correspondence: dtembo@mubas.ac.mw

**Abstract:** This study evaluated the phytochemical composition and antioxidant activity of *Piliostigma thonningii* (Schumach.) Milne-Redh, *Psorospermum febrifugum* Spach, *Inula glomerata* Oliv. and Hiern, *Zanthoxylum chalybeum* Engl. and *Monotes africanus* A.DC., claimed to treat cancer by Malawian traditional herbal practitioners. Ground and dried plant extracts were analyzed for total phenolic content (TPC), total flavonoid content (TFC), total alkaloid content (TAC), ferric reducing antioxidant power (FRAP) and 2,2-Diphenyl-1-Picrylhydrazyl (DPPH) using standard assays. The TPC, TFC, and TAC ranged from $539 \pm 2.70$ to $4602 \pm 32$ mg GAE/g DW, $6.18 \pm 0.03$ to $64.04 \pm 0.16$ mg QE/g DW and $19.25 \pm 0.07$ to $76.05 \pm 0.36$ mg CE/g DW, respectively, and the variations were significant, $p < 0.05$. FRAP values ranged from $82.15 \pm 0.7$ to $687.28 \pm 0.71$ mg TEAC/g DW and decreased in the following order: *P. thoningii* (Schumach.) Milne-Redh > *P. febrifugum* Spach > *M. africanus* A.DC > *Z. chalybeum* Engl > *I. glomerata* Oliv. and Hiern. The scavenging activity ($SA_{50}$) of the extracts ranged from $0.09 \pm 0.01$ to $1.57 \pm 0.01$ μg/mL of extract with *P. thonningii* (Schumach.) Milne-Redh showing the lowest value. Based on the levels of phenolic compounds and their antioxidant activity, the plants in this study could be considered for use as medicinal agents and sources of natural bioactive compounds and antioxidants.

**Keywords:** antioxidant activity; medicinal plants; phytochemicals; anticancer properties; traditional herbal practitioners

## 1. Introduction

Natural compounds from some plants have anticancer properties with lower toxicity properties. These phytochemicals act as antioxidants by scavenging free radicals that are produced in the body. They also act as anti-inflammatory and anticancer agents by suppressing or blocking cancerous cell pathways [1]. The known anticancer phytochemicals in plants include phenolics (including flavonoids) and alkaloids [2,3]. The overproduction of free radicals (oxidants) can cause an imbalance, leading to oxidative stress, with subsequent oxidative damage to large biomolecules such as lipids, proteins, and deoxyribonucleic acids (DNA), resulting in an increased risk of cancer [2,3]. Natural antioxidants in plants are thought to inhibit free radical chain reactions in the body by preventing initiation or propagation steps, causing chain termination reactions, and thereby delaying the oxidation process [4]. Free radical species such as superoxide ($O_2^{\bullet-}$), hydroxyl ($OH^-$), and nitric oxide ($^{\bullet}NO$) are generated in the body during normal cellular metabolism, and their normal concentration in the body is maintained [5]. At normal levels, free radicals enact useful

normal physiological protective mechanisms. Nevertheless, when the reactive oxygen species are overproduced or the antioxidant system has been compromised, oxidative stress occurs [6–9]. When in excess, these free radicals damage macromolecules such as deoxyribonucleic acid (DNA), cellular proteins, and unsaturated fatty acids, impairing the macromolecules' proper functioning and resulting in degenerative human diseases, such as cancer [5,10].

Plants contain antioxidant secondary metabolites such as phenolics, flavonoids, alkaloids and ascorbic acid [11]. These antioxidants are strong scavengers of free radicals in the body, thereby averting oxidative stress damage to cellular components [11–13]. In addition, antioxidants have both preventive and curative pharmacological activities against a wide range of diseases, including diabetes, cancer, inflammation, and dementia [14–17]. Despite the availability of many synthetic drugs used to manage oxidative stress, the high costs and adverse side effects associated with them limit their usefulness [18]. As a result, alternative nontoxic antioxidants, which are affordable, are needed to counter oxidative stress, thereby thwarting the associated diseases [19]. Plants have phenolic and alkaloid compounds that have been shown to have an array of in vitro and in vivo antioxidant effects [20,21].

Many Malawian traditional herbal practitioners (THP) claim to know of medicinal plants with antioxidant activities, and use such medicinal plants for cancer treatment and management [22,23]. In the northern region of Malawi, especially in Nkhata Bay and Mzimba districts, THPs use root barks of *P. thonningii* (Schumach.) Milne-Redh (Monkey bread)**,** *P. febrifugum* Spach (Christmas berry) and *I. glomerata* Oliv. and Hiern (Hare's ears), and stem barks of *Z. chalybeum* Engl. (Knob wood) and *M. africanus* A.DC (Pink-fruited monotes), to treat and manage unhealing wounds, prostate cancer, cervical cancer, and stomach ulcers.

*P. thonningii* (Schumach.) Milne-Redh belongs to the Fabaceae family, and is usually a small- to medium-sized rounded tree, 3–5 m high, but it may reach 10 m in ideal conditions. *P. thonningii* is traditionally used for the management of inflammation, malaria, fever, rheumatism, and mental illness, among other diseases caused by a disturbed redox state in the body [5]. In addition, Alagbe [20] reported that the leaves, roots, and stem bark have been traditionally used for the treatment of chronic ulcers, diarrhea, toothache, gingivitis, cough, bronchitis, snake bites, hookworms and skin diseases *P. febrifugum* Spach belongs to the Hypericaceae family. It is a shrub or small tree, 3–4 m high by occasionally reaching 7 m, occurring over a wide range of altitudes and scattered through open woodland. The stem bark of *P. febrifugum* Spach from Cameroon also has shown antitumor, anticancer, and antioxidant activities, while traditional medicine practitioners in Uganda use it for the treatment of skin sores in HIV/AIDS patients [24]. *I. glomerata* Oliv. and Hiern of the Asteraceae family is a robust perennial herb, which grows up to 1.5 m high, with basal rosette leaves showing an irregularly toothed margin. Its roots are used to treat hypertension, while its leaves are used for treating erectile dysfunctions [8]. *Z. chalybeum* Engl belongs to the Rutaceae family. In Uganda, *Z. chalybeum* Engl. is used for treating tuberculosis, malaria and sickle cells, and the root or stem barks are the most important sources of medicine [25]. *M. africanus* A.DC belongs to the Dipterocarpaceae family, and is usually a small tree of 8 m high with simple concolorous leaves. *M. africanus* A.DC is reported to have anti-HIV effects [26].

In Malawi, most herbal plant species are promoted as medicinal plants without scientific evidence, and little work has been done to evaluate and validate their effectiveness [27]. To the best of our knowledge, there is no scientific study on the antioxidant activities, total phenolic, flavonoid, and alkaloid contents of the five plants from Mzimba and Nkhata Bay districts. This study was, therefore, designed to evaluate and validate the in vitro ferric reducing antioxidant power (FRAP) and 2,2-Diphenyl-1-Picrylhydrazyl (DPPH) antioxidant capacities, including total phenolic, flavonoid and alkaloid contents, of root barks of *P. thonningii* (Schumach) Milne-Redh, root barks of *P. febrifugum* Spach, leaves of *I. glomerata* Oliv. and Hiern, stem barks of *Z. chalybeum* Engl. and leaves of *M. africanus* A.DC, which

are used by traditional herbal practitioners to treat and manage cancer in the Mzimba and Nkhata Bay districts of North Malawi.

## 2. Materials and Methods

### 2.1. Chemicals

Chemicals such as ascorbic acid ((R)-3,4-dihydroxy-5-((S)-1,2-dihydroxyethyl)furan-2(5H)-one), trolox (6-hydroxy-2,5,7,8-tetramethylchromane-2-carboxylic acid), gallic acid (3,4,5-trihydroxybenzoic acid), quercetin dehydrate (3,5,7-trihydroxy-2-(3,4-dihydroxyphenyl)-4H-chromen-4-one dihydrate) and caffeine (1,3,7-trimethyl-1H-purine-2,6(3H,7H)-dione) were used as standards and were of analytical reagent (AR) grade. A solution of 98% *v/v* sulfuric acid, anhydrous sodium sulfate, ammonium molybdate, citric acid monohydrate, glacial acetic acid, sodium acetate trihydrate, iron (III) chloride hexahydrate, aluminum (III) chloride, di-sodium orthophosphate ($Na_2HPO_4$), citric acid monohydrate, anhydrous sodium carbonate, 32% *v/v* hydrochloric acid, sodium acetate trihydrate, sodium hydroxide pellets, bromocresol green, and chloroform was purchased from Saarchem (Johannesburg, RSA); Folin–Ciocalteu phenol reagent and 2,4,6-tris-2-pyridyl-s-triazine (TPTZ) were purchased from Sigma (Burlington, MA, USA), while 2,2-diphenyl-1-picrylhydrazyl (DPPH) was purchased from Sicco Research laboratories (Mumbai, India). Trolox was purchased from Calbiochem (Darmstadt, Germany); quercetin dihydrate was purchased from EMD Millipore (Billerica, MA, USA), while caffeine was purchased from BDH Chemicals (Poole, UK).

### 2.2. Plant Materials

The leaf, root and stem samples were sustainably harvested in October, 2020, from the Mzimba and Nkhata Bay districts, North Malawi. The five plants were found at the following Global Positioning System (GPS) coordinates: *P. thonningii* (Schumach.) Milne-Redh (E: 0598133, N: 8765558; Elevation: 1093), *P. febrifugum* Spach (E: 0585012, N: 8737972; Elevation: 1208), *I. glomerata* Oliv. and Hiern (E: 0584798, N: 8738115; Elevation: 1196), *Z. chalybeum* Engl. (E: 0585321, N: 8737939; Elevation: 1230) and *M. africanus* A.DC (E: 0607717, N: 8739023; Elevation: 1305). The plants were identified and authenticated by a taxonomist from the National Herbarium and Botanical Gardens (NHBG) of Malawi, Mzuzu Office. The five plants were assigned the following specimen numbers: MLW-FM-MZ/ENU-001, MLW-FM-MZ/MTW-007, MLW-FM-MZ/MTW-009, MLW-FM-MZ/LUP-003 and MLW-FM-MZ/MTW-002 for *P. thonningii* (Schumach.) Milne-Redh, *P. febrifugum* Spach, *I. glomerata* Oliv. and Hiern, *Z. chalybeum* Engl. and *M. africanus* A.DC, respectively. Finally, the specimens along with their voucher numbers were deposited at the NHBG of Malawi in Mzuzu.

### 2.3. Sample Preparation

Samples were washed with tap water to remove any dirt and soil, as previously described by Imad et al. [28]. Root and stem barks were cut into smaller pieces to enhance drying. Leaf samples were not cut to smaller pieces. The samples were sorted, named accordingly, and shed-dried as described by Nantongo et al. [25] for one month in the chemistry laboratory. After one month of shed-drying, the samples were pulverized using a Huang Cheng Yan high speed multifunctional mill (CGOLDENWALL), sieved through a 0.25 mm-mesh size using sieve number 60 and transferred into sealed bottles. The sealed bottles containing powdered samples were transferred into black plastic bags and kept in the dark until analyses.

### 2.4. Moisture Content

The percent moisture of the pulverized and sieved plant samples was determined using the method described by Tembo et al. [29]. Samples (2 g) were accurately weighed in triplicate in labelled, preheated, desiccator-cooled, and pre-weighed porcelain crucibles with covers on a PW-214 AE Adams analytical balance (Isando, RSA). The samples in the covered porcelain crucibles were then placed in a Gallenkamp Pius II hot air oven (Cam-

bridge, UK), set and thermostatically controlled at 110 °C overnight (12 h), during which the samples dried to a constant mass. The results are presented as percent moisture content.

## 2.5. Extraction of Phytochemicals

Extractions of phytochemicals were undertaken as described in the literature [23,30–33]. Twenty percent mass per volume (20% *m/v*) mixtures were prepared by weighing the pulverized plant samples (20 g) into 250 mL quick fit Erlenmeyer flasks followed by the addition of 80% *v/v* methanol (100 mL), and stoppered. The plant and methanol mixtures were then magnetically stirred (Labcon MH10, Chamdor RSA) at a moderate speed for 2 h at an ambient temperature. Thereafter, the mixtures were transferred into 50 mL falcon tubes, vortexed (VarMix Vortex by SciQuip, Stuttgart, Germany) for 1 min, centrifuged (Thermo Scientific Medifuge centrifuge, Karlsruhe, Germany) at 4000 revolutions per minute (rpm) for 10 min followed by gravity filtration using Whatman filter paper No. 1. The residue was re-extracted with a second 80% *v/v* methanol (100 mL) and the two filtrates were pooled together into a pre-weighed quick fit round-bottomed flask. The solvent of the crude plant extracts was evaporated *in vacuo* using a rotary evaporator (BUCHI R100 Labortechnik AG, Flawil, Switzerland). The semidried residue was quantitatively transferred into 100 mL plastic beakers and further dried to a constant mass using a water bath (Clifton NE 2-28D by Nickel-Electron Limited, Weston-super-Mare, UK) set at 40 °C. The dried sample was weighed, transferred into sealed sample tubes, kept in a black plastic bag, and stored under refrigeration at 4 °C.

## 2.6. Preparation of 10 mg/mL and 1 mg/mL Stock and Working Plant Extracts, Respectively

Here, 10 mg/mL stock extract solutions were prepared by weighing and dissolving the dried extracts (0.1 g) into 50 mL falcon tubes followed by the addition of 80% methanol (MeOH) (10 mL) using a Lab bottle top dispenser (Shangai Rongtai Biochemical Company, Shanghai, China). In this way, 80% methanol is able to extract 100% of the phenolic compounds, some of which are more water-soluble (hydrophilic) [34]. In addition, polar phytochemicals are present as dipoles, and they interact with one another electrostatically in solid form. Polar solvents also interact with the dipolar phytochemicals, and such interactions weaken the bonds between solid phytochemicals, resulting in enhanced dissolving [35]. In addition, 80% *v/v* methanol has more polar organic properties, and represents a better solvent for the polar organic phytochemicals [32]. When in solution, the dipolar phytochemicals are solvated (surrounded) by the polar solvents, and consequently keep in solution to stop the dipolar phytochemicals from recombining [35]. The mixtures were vortexed for 1 min, then centrifuged at 4000 rpm for 10 min, and gravity-filtered using Whatman No.1 filter paper into 15 mL falcon tubes, which were then sealed. In total, 1 mg/mL working plant extract solutions (10 mL) were prepared by pipetting and diluting 1 mL of the 10 mg/mL stock plant extracts into 10 mL volumetric flasks, filling to the mark with 80% *v/v* methanol, then stoppering and homogenizing. Both the stock and working plant extract solutions were stored under refrigeration at 4 °C till subsequent analyses.

## 2.7. Determination of FRAP and DPPH Antioxidant Activities

The ferric reducing antioxidant power (FRAP) was determined as described by some researchers [25,31,36,37]. Standards of trolox ranging from 0 to 100 mg/100 mL were prepared, and solutions of both the standards and 1 mg/mL sample extracts (1 mL) were pipetted into 50 mL falcon tubes using a 1000 μL Eppendorf micro-pipette followed by the addition of FRAP reagent (6 mL) using a Lab bottle top dispenser. The mixtures were vortexed for 1 min and incubated at ambient temperature for 10 min. After the 10 min incubation period, the samples were transferred into 10 mm cuvettes and their absorbance read at 593 nm using a UV-Vis spectrophotometer (Spectro 2092 PLUS, Analytical Technologies Limited, Gujarat, India). FRAP antioxidant activity was determined in triplicate and expressed as mg trolox equivalent antioxidant capacity (TEAC)/g dry weight (DW). In total, 20 μg/mL of dried plant extract (1 mL) was prepared by diluting the 1 mg/mL crude

extract (0.02 mL) with 80% *v/v* MeOH (0.08 mL) in 50 mL falcon tubes using a 10–1000 μL Eppendorf micro-pipette. The DPPH antioxidant activities of the 20 μg/mL MeOH extracts' were determined using a 0.1 mM DPPH assay as described by Molyneux [38] and Masalu et al. [39], with some modifications. Later, 0.1 mM of DPPH solution (4.0 mL) was added to the mixtures in falcon tubes using a Lab bottle top dispenser. The volumes of both 80% *v/v* methanol (1.0 mL) and trolox (20 μg/mL, 1 mL) served as negative and positive controls, respectively, and were similarly treated with a 0.1 mM DPPH solution (4.0 mL). The mixtures of both extracts and controls were then vortexed for 30 s and allowed to stand in the dark at ambient temperature for 30 min. Absorbance values of the resulting solutions were measured at 517 nm using a Spectro 2092 PLUS UV/Vis spectrophotometer.

$$\text{Percentage (\%) DPPH scavenging activity} = (1 - (A_s/A_c) \times 100),$$

where $A_s$ is the absorbance of the sample while $A_c$ is the absorbance of the blank (control). A standard calibration plot was used to calculate the concentration of the extract that would halve the scavenging activity of 0.1 mM DPPH solution. $SA_{50}$ is the concentration in μg/mL of the plant extract required to scavenge 50% of 0.1 mM DPPH, according to Masalu et al. [39].

### 2.8. Total Phenolic, Flavonoid, and Alkaloid Contents

Total phenolic content (TPC) was determined using the Folin–Ciocalteu (FC) assay as previously described [23,29,31]. Standards of gallic acid ranging from 0 to 100 mg/L and blank (80% *v/v* methanol) were prepared. Aliquots of both standards, 1 mg/mL extracts, and blanks (1 mL) were transferred into 15 mL falcon tubes using an Eppendorf micro-pipette followed by the addition of 10-fold diluted FC reagent (5 mL) and 1 M sodium carbonate (4 mL) using a sample dispenser. The preparation of the samples and reagents was done within 3–8 min, followed by vortexing for 1 min, and left to stand for 2 h to allow color development. The samples were then transferred into 10 mm cuvettes and their absorbance read at 765 nm using a UV-Vis spectrophotometer. The TPC analysis was done in triplicate, and the results are expressed as milligram of gallic acid equivalents per gram of dry weight (mg GAE $g^{-1}$ DW).

The total flavonoid content (TFC) was determined using the aluminum chloride colorimetric method as described by Mwamatope et al. [23] and Santos et al. [31]. Samples of both quercetin standards, 80% *v/v* MeOH (blank), and 1 mg/mL extracts (2 mL) were pipetted into 15 mL falcon tubes followed by the addition of 2% aluminum (III) chloride (A$\ell$C$\ell_3$) (2 mL) using an Eppendorf micro-pipette. The mixtures in the falcon tubes were vortexed for 1 min and incubated at ambient conditions for 30 min. After the incubation period, their absorbance values were read at 415 nm using 10 mm cuvettes and a UV-Vis spectrophotometer. The TFC analysis was done in triplicate, and the results were expressed as milligram of quercetin equivalent per gram of dry weight (mg QE $g^{-1}$ DW).

Total alkaloid content (TAC) was estimated photometrically using the bromocresol green (BCG) method, as described by several authors [25,31,36,38]. The BCG assay is based on the formation of a yellow-colored complex formed from a reaction between BCG and alkaloids. Caffeine working solutions of 0–2 μg/mL were prepared from a 100 μg/mL stock solution. Dried plant extract samples (0.1000 g) were weighed using a PW-214 AE Adams analytical balance (RSA) into 15 mL falcon tubes followed by the addition of 2N hydrochloric acid solution (5 mL) to dissolve the sample. The mixtures were then vortexed for 2 min, followed by centrifuging at 4000 rpm for 10 min. Volumes of each extract, including the working standards (1.0 mL), were transferred into 50 mL falcon tubes using an Eppendorf micro-pipette followed by the addition of phosphate buffer (5 mL) and BCG (5 mL). The mixtures were vortexed for 1 min using VarMix Vortexer. Chloroform (CHCl$_3$) (5 mL) was then added to the mixtures, which were swirled to allow the yellow complex to separate in the CHCl$_3$ layer. After phase separation, the upper yellow CHCl$_3$ layer was pipetted into a 10 mL volumetric flask using a Pasteur pipette and filled to the mark with CHCl$_3$. The yellow complex solution was transferred into a 10 mm silica cuvette and the

absorbance was read at 450 nm against a blank ($CHCl_3$) using a UV-Vis spectrophotometer. The TAC analysis was done in triplicate, and the results are expressed as milligram of caffeine equivalent per gram of dried weight (mg CE $g^{-1}$ DW) of sample.

### 2.9. Statistical Analysis

The analyses were done in triplicate and the data obtained are expressed as mean $\pm$ standard error of the means (mean $\pm$ S.E.M). The data have been subjected to one-way analysis of variance (ANOVA) and significance has been declared if $p \leq 0.05$.

## 3. Results

### 3.1. Moisture Content (%)

The percent moisture contents of the pulverized and sieved plant samples ranged from 14.57% to 17.62% (Table 1).

**Table 1.** Percent yield of crude plant extracts of the five plants.

| Plant | *P. thonningii* (Schumach) Milne-Redh | *P. febrifugum* Spach | *I. glomerata* Oliv. and Hiern | *Z. chalybeum* Eng. | *M. africanus* A.DC |
|---|---|---|---|---|---|
| % Moisture | 17.62 $\pm$ 0.21 [a] | 16.67 $\pm$ 0.14 [b] | 17.40 $\pm$ 0.19 [a] | 14.57 $\pm$0.11 [d] | 15.72 $\pm$ 0.13 [c] |
| % Yield | 49 $\pm$ 0.15% | 45 $\pm$ 0.10% | 6 $\pm$ 0.03% | 20 $\pm$ 0.05% | 19 $\pm$ 0.03% |
| Reference | 39% [5] | 30.8% [24] | 8.5% [8] | N/A | N/A |

N/A: not accessed, $n$ = 3, mean $\pm$ standard deviation. Values with different superscripts are significantly different ($p < 0.05$).

### 3.2. Yield (%) of the Crude Plant Extracts

The percent yields of the crude plant extracts were highest in *P. thonningii* (Schumach) Milne-Redh (49%), followed by *P. febrifugum* Spach (45%), then 20% for *Z. chalybeum* Eng. and 19% for *M. africanus* A.DC, while *I. glomerata* Oliv. and Hiern yielded 6%. (Table 1).

### 3.3. FRAP and DPPH Antioxidant Activities of the Plants

In this study, the FRAP antioxidant activity ranged from 82.15 $\pm$ 0.7 to 687.28 $\pm$ 0.71 mg TEAC/g DW (Table 2). The FRAP antioxidant activity decreased in the following order: *P. thoningii* > *P. febrifugum* > *M. africanus* > *Z. chalybeum* > *I. glomerata*. The $SA_{50}$ results in the current study ranged from 0.09 $\pm$ 0.01 to 1.57 $\pm$ 0.01 $\mu$g/mL of extract, while that of the positive control (trolox) was 0.05 $\mu$g/mL.

**Table 2.** Antioxidant activities of the medicinal plant species.

| Medicinal Plants | FRAP (mg TEAC/g DW) | $SA_{50}$ (DPPH) ($\mu$g/mL) |
|---|---|---|
| *P. thonningii* (Schumach.) Milne-Redh | 687.28 $\pm$ 0.71 [a] | 0.09 $\pm$ 0.01 [a] |
| *P. febrifugum* Spach | 401.11 $\pm$ 0.41 [b] | 0.21 $\pm$ 0.01 [a] |
| *I. glomerata* Oliv. and Hiern | 120.23 $\pm$ 0.12 [e] | 0.14 $\pm$ 0.01 [a] |
| *Z. chalybeum* Engl. | 82.15 $\pm$ 0.07 [c] | 1.57 $\pm$ 0.01 [c] |
| *M. africanus* A.DC | 123.86 $\pm$ 0.14 [d] | 1.29 $\pm$ 0.02 [b] |
| Trolox | | 0.05 $\pm$ 0.01 |

$n$ = 3 and values with different superscripts are significantly different ($p < 0.05$).

### 3.4. Total Phenolic, Flavonoid, and Alkaloid Contents

3.4.1. Total Phenolic Content (TPC)

The results in Figure 1 indicate the variation in TPC for different types of plant samples. The TPC contents ranged from 539 $\pm$ 0 mg to 4602 $\pm$ 32 mg GAE/g DW.

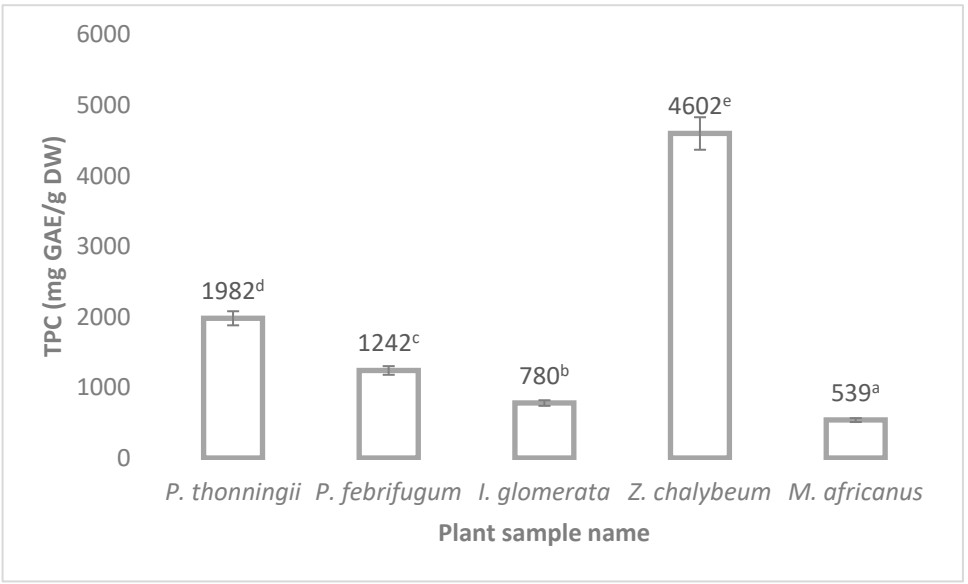

**Figure 1.** Total phenolic contents (mg GAE g$^{-1}$ DW) of medicinal plants. Mean values that do not share a letter indicate significant differences ($p < 0.05$).

3.4.2. Total Flavonoid Content (TFC)

TFC values for the five plants are given in Figure 2. The TFC results range from 6.18 ± 0.00 to 64.04 ± 0.16 mg QE/g DW, in the following (increasing) order: *Z. chalybeum < P. thonningii < I. glomerata < M. africanus < P. febrifugum*.

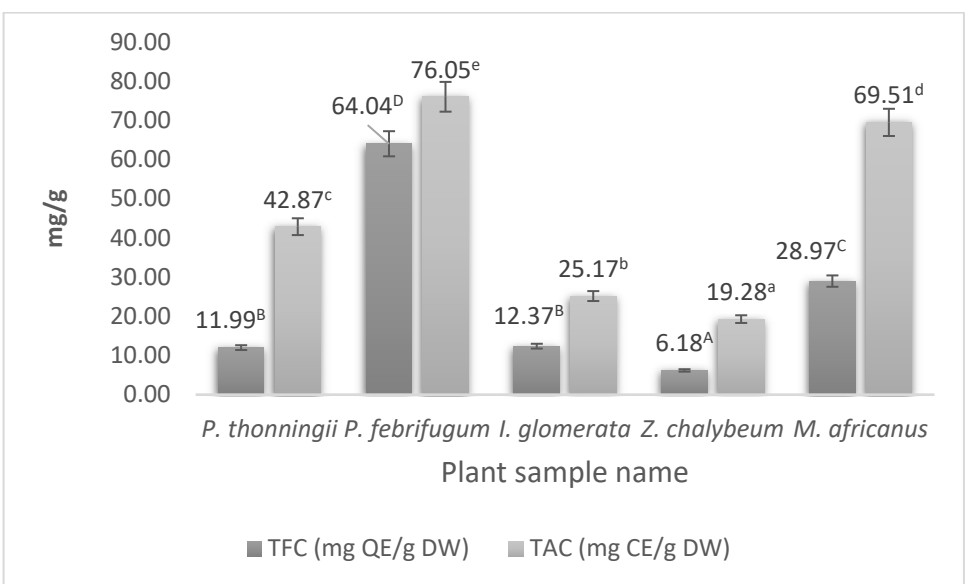

**Figure 2.** Total flavonoids content (mg QE g$^{-1}$ DW), and total alkaloids (mg CE g$^{-1}$ DW), of medicinal plants. Mean values (capital letters for TFC and small letters for TAC) that do not share a letter indicate significant differences ($p < 0.05$).

3.4.3. Total Alkaloid Content (TAC)

Alkaloids possess analgesic, antibacterial, and antiplasmodic properties [40], and the results for this content shown in Figure 2 range from 19.25 ± 0.07 to 76.05 ± 0.36 mg CE/g DW. Their increasing order is as follows: *Z. chalybeum < I. glomerata < P. thonningii < M. africanus < P. febrifugum*.

## 4. Discussion

The moisture content of the pulverized plant samples ranged from 14.57% to 17.62%. The moisture content for *P. thonningii* (Schumach) Milne-Redh obtained in our study was higher than the 8.34% reported by Alagbe [20]. The variability of moisture content in the samples could be due to the uncontrolled drying associated with shed-drying [41,42]. The phytochemical yields obtained depend on the ages of plants, drying processes, extraction methods, geographical locations and soil types [20,21,23,25,43]. The yields of dried crude plant extracts were highest in *P. thonningii* (Schumach) Milne-Redh and *P. febrifugum* Spach. The 49% yield for *P. thonningii* (Schumach) obtained in our study was higher than the 39% reported by Moriasi et al. [5]. Moriasi et al. [5] used pure methanol during extraction. Pure methanol may not have extracted all the hydrophilic phytochemicals, as reported by Chigayo et al. [32] and Che Sulaiman et al. [34]. The 45% dried extract from *P. febrifugum* Spach was higher than the 30.8% yield reported by Konan et al. [24]. However, Konan et al. [24] used similar extraction conditions as our study. Therefore, the lower percent yield could be due to differences in the ages of plants, the drying processes, the geographical locations or the soil types [20,21,23,43]. Finally, the 6% yield of *I. glomerata* Oliv. and Hiern was lower than the 8.5% reported by Ojo et al. [8], who used 17% *v/v* methanol as a solvent. According to Chigayo et al. [32] and Che Sulaiman et al. [34], solvents with lower than 80% *v/v* methanol content may not be as polar, preventing them from extracting most of the polar organic phytochemicals. Therefore, the higher percent yield obtained by Ojo et al. [8] could not be due to the low (17% *v/v*) methanol content of the solvent. The higher yield obtained could be due to differences in ages of plants, drying processes, geographical locations or soil types [20,21,23,43]. This means that factors such as the ages of plants, drying processes, extraction methods, geographical locations and soil type should be considered when using plants as herbal medicines.

The FRAP antioxidant activity results ranged from $82.15 \pm 0.7$ to $687.28 \pm 0.71$ mg TEAC/g DW, and were within the 40.00 to 31,050 mg TEAC g$^{-1}$ DW range reported by Surveswaran et al. [44]. Similar observations of relatively high FRAP values in medicinal plants have been previously reported [45]. $SA_{50}$ is defined as the concentration of total antioxidant necessary to reduce the initial radical concentration of DPPH by 50% [39]. The decrease in concentration is also accompanied by a proportionate decrease in absorbance, as per the Beer–Lambert law. *P. thonningii* (Schumach) Milne-Redh had the highest value, followed by *P. febrifugum* Spach, in terms of both FRAP and DPPH antioxidant activities. The studied plants had relatively high FRAP values. Medicinal plants with considerably high antioxidant activity have been reported to possess various biological and pharmacological properties [6,16,18]. However, confirmatory investigations of such activities are needed for the medicinal plants under study. FRAP is a single electron transfer (SET)-based assay [46,47], while DPPH, by virtue of being a free radical, undergoes a hydrogen atom transfer (HAT) mechanism that enables the hydrogen atom to bring the electron required for the formation of a single covalent bond [48]. Therefore, of the studied plants, *P. thonningii* (Schumach) Milne-Redh had the highest levels of antioxidants, which can scavenge endorgenic free radicals (pro-oxidants) through both SET and HAT mechanisms (Table 2). It should, however, be noted that high DPPH values could also be due to the presence of non-phenolic antioxidants, which may also quench endogernic free radicals [49]. However, *Z. chalybeum* Engl. and *I, glomerata* Oliv. and Hiern had the lowest HAT- and SET-based antioxidants levels, respectively (Table 2). The low $SA_{50}$ results of the plant extracts imply that the studied plants are strong in vitro scavengers of the DPPH radical. The strong antioxidant activities could be attributed to the presence of bioactive antioxidant phytochemicals in these extracts, which work synergistically to scavenge the DPPH radicals [11]. The antioxidant activity results suggest that all five of the studied plants in this study could potentially restore and modulate the activity of endogenous antioxidant systems. Similarly, this supports the findings of earlier studies by Santos et al. [31], Zhang et al. [16] and Moriasi et al. [5]. Therefore, the root barks of *P. thonningii* (Schumach) Milne-Redh, root barks of *P. febrifugum* Spach, leaves of *I. glomerata* Oliv. and Hiern, stem barks of *Z. chalybeum* Engl.

and leaves of *M. africanus* A.DC can attenuate the damaging effects caused by oxidative stress. However, further studies will be needed to analyze in vivo anticancer activity using cell lines and the fingerprinting of specific anticancer phytochemical properties.

Phenolic acids are derivatives of benzoic or cinnamic acids, which form hydroxybenzoic and hydroxycinnamic acids, respectively. These phytochemicals contribute significantly to the antioxidant properties of plant extracts [24], which are capable of scavenging free radicals and consequently preventing diseases [50]. The results in Figure 1 indicate the variation in TPC for different types of plant samples. The observed variations could be attributed to the differences in genetic composition, geographical location, environmental conditions, stage of maturity and soil type [20,21,23,43,51]. A total phenolic content of 50.2 mg GAE/g DW for *P. thonningii* (Schumach) Milne-Redh was reported by Alagbe [20], which result is less than the $1982 \pm 2$ mg GAE/g DW value obtained in this study. In addition, Ojo et al. [8] reported a TPC value of 0.08 mg GAE/g DW for *I. glomerata* Oliv. and Hiern, which is also lower than the $780 \pm 4$ mg GAE/g DW obtained in this study. Furthermore, Nantongo et al. [25] reported a TPC value of 1.70 mg GAE/g DW for *Z. chalybeum* Engl. stem bark, which is also lower than $4602 \pm 32$ mg GAE/g DW. During extractions, Alagbe [20] used diethyl ether, while Ojo et al. [8] and Nantongo et al. [25] used commercial-grade methanol with no water added. The usage of solvents that are so different from the 80% *v/v* used in our study might have contributed to the low TPC values, as diethyl ether is less polar than 80% *v/v* methanol. In addition, the pure methanol used by Nantongo et al. [25] during extraction may not have extracted most of the polar organic phytochemicals, as reported by Che Sulaiman et al. [34]. However, a TPC value of 3761 mg/GAE/g DW reported by Alsiede [52] was derived from a dried extract fraction obtained via a sequential extraction procedure. Alsiede initially defatted the powdered *Cassia singueana* samples using petroleum ether (60–80 °C), followed by sequential extraction using chloroform, ethyl acetate, and finally methanol. The TPC values obtained in our study were from crude extracts. The fraction yields obtained from sequential extractions would be lower than those from crude extracts. Therefore, such a high value of TPC obtained by Alsiede [52] might have been due to differences in genetic composition, the age of plants, the drying processes, the extraction methods, the geographical location and the soil type [20,21,23,25,43].

Flavonoids have antifungal, antibacterial, and antioxidant properties [53,54]. The TFC result of $11.99 \pm 0.0.01$ mg QE/g DW for *P. thonningii* (Schumach) Milne-Redh root bark obtained in our study is lower than both the 35.0 mg QE/g DW reported in India by Alagbe [20] and the 52.3 mg QE/g DW reported in Burkina Faso by Sombie et al. [49]. As indicated earlier, Alagbe [20] used diethyl ether as a solvent. Unless the *P. thonningii* (Schumach) Milne-Redh used had a moderately high polar TFC, it is doubtful whether the less polar diethyl ether would have positively contributed to the yield of the TFC, because Chigayo et al. [32] reported that diethyl ether extracts usually contain low yields as compared to more polar extracts, such as methanol and water. The difference in TFC content between our result of $11.99 \pm 0.0.01$ mg QE/g DW and the 35.0 mg QE/g DW reported by Alagbe [20] could therefore be attributed to differences in geographical location, stage of maturity, drying processes, extraction processes and soil type [20,21,23,43]. Sombie et al. [49] used an aqueous decoction with an unspecified temperature of extraction. Elevated decoction temperatures of $\geq 60$ °C and less than 80 °C are reported to maximize extraction yields [34]. Therefore, the high yield of 52.3 mg QE/g DW reported by Sombie et al. [49] could be related to the elevated decoction temperature that was used.

Alkaloids possess analgesic, antibacterial, and antiplasmodic properties [40]. The TAC of $19.28 \pm 0.01$ mg CE/g DW obtained from *Z. chalybeum* Engl. stem bark was higher than the 0.08 mg CE/g DW reported by Nantongo et al. [25], but lower than the 71.3 mg CE/g DW reported by Alagbe [20]. Nantongo et al. [25] used pure commercial methanol, while Alagbe [20] used diethyl ether as the solvent. The low TAC yield obtained in Nantongo's work may have been contributed by the pure methanol used, since pure alkanols are not as efficient in extracting polar compounds such as alkaloids [32]. The high TAC yield

obtained from the diethyl extract and reported by Alagbe [20] may be due to other factors, such as differences in the geographical locations, ages of plants, drying processes and soil types [20,21,23,43,52]. This could be the case since moderately polar solvents such as diethyl ether are less efficient in extracting polar solutes such as alkaloids [32,34].

Total phenolic contents are usually higher than the flavonoid contents [55]. This is expected because flavonoids are a subclass of phenolics. In most plants, the common order of secondary metabolites is phenolics > alkaloids > flavonoids [25]. Both trends have been maintained in our results, as demonstrated in Figures 1 and 2. One of the factors influencing the distribution of phytochemicals within a plant is environmental conditions. The *Z. chalybeum* Engl. was harvested from an anthill within a thick forest. The abundance of total phenolics was the highest in *Z. chalybeum* Engl., while this showed the lowest levels of alkaloids. Differences in metabolite abundance have been detected among and within species primarily due to genetic factors, environmental effects and their interaction [25,56–58]. Changing growth conditions, especially nitrogen (N) availability, have been shown to affect phenolic concentrations in plant tissues. Specifically, N deficiency or limitation leads to phenolic accumulation in different plant parts, such as stems and roots [57,58]. The comparatively higher levels of constitutive secondary metabolites observed in *Z. chalybeum* Engl. may also reflect the levels of biotic and abiotic stress it experiences [59]. These stresses are typical of the natural forests where the *Z. chalybeum* Engl. samples were collected.

## 5. Conclusions

Root barks of *P. thonningii* (Schumach) Milne-Redhead, root barks of *P. febrifugum* Spach, leaves of *I. glomerata* Oliv. and Hiern, stem barks of *Z. chalybeum* Engl. and leaves of *M. africanus* A.DC had strong FRAP and DPPH antioxidant activities. In addition, the same plants had phenolics, including flavonoids and alkaloids, suggesting that they could play an important role in preventing and managing many health problems, such as cancer, cardiovascular diseases, diabetes, and obesity. These plants should, however, be further analyzed for their in vivo anticancer activity using cell lines. In addition, a further study leading to the fingerprinting of specific anticancer phytochemicals is recommended.

**Author Contributions:** Conceptualization, F.F.F.M. and D.T.; methodology, F.F.F.M. and D.T.; validation, F.F.F.M., J.F.K., A.M. and D.T.; formal analysis, F.F.F.M.; investigation, F.F.F.M.; resources, F.F.F.M., B.M. and D.T.; data curation, F.F.F.M. and B.M.; writing—original draft preparation, F.F.F.M.; writing—review and editing, F.F.F.M., J.F.K., A.M., B.M. and D.T.; visualization, supervision and project administration, F.F.F.M., J.F.K., A.M., B.M. and D.T.; funding acquisition, F.F.F.M. and J.F.K. All authors have read and agreed to the published version of the manuscript.

**Funding:** The research study was funded by the Kamuzu University of Health Sciences with funding from the African Centre of Excellence in Public Health and Herbal Medicine (ACEPHEM), Project ID number P151847.

**Institutional Review Board Statement:** Not applicable.

**Informed Consent Statement:** Not applicable.

**Data Availability Statement:** The data presented in this study are available on request from the corresponding author.

**Acknowledgments:** The authors wish to acknowledge and thank Kamuzu University of Health Sciences and Mzuzu University for providing laboratory space. We also acknowledge the support given by the president of Northern Region Traditional Healers Association of Malawi and the botanist from the National Herbarium and Botanical Gardens of Malawi, Mzuzu office.

**Conflicts of Interest:** The authors declare that there are no conflict of interest associated with this work and the publication of this paper.

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
