# Peer review of "Antioxidant Activity of Selected Medicinal Plants Used by Traditional Herbal Practitioners to Treat Cancer in Malawi"

_2571-8800, doi:10.3390/j6040039_

Round 1

Reviewer 1 Report

The authors have submitted a manuscript in which they evaluate for the first time the in vitro antioxidant capacities of stem barks of P. thonningii (Schumach) Milne-Redhead, P. febrifugum Spach, I. glomerata Oliv. & Hiern and root barks of Z. chalybeum Engl. and M. africana A.D.C from Mzimba and Nkhata Bay districts.

The title does not reflect the real purpose of the work. Furthermore, the Introduction is too long and dispersive; it does not reveal the true content of the article and the novelty is not well explained. Reading the entire paper, it seems that the paper deals more with the characterization of the antioxidant properties of these plants rather than their anti-cancer potential. The link between antioxidant and anticancer potential is too tenuous to warrant such work. Anti-cancer activity on cell lines is advisable to enrich this paper in order to keep this focus.

Paragraph 2.4: Has the temperature been controlled or have they been dried in the shade at room temperature? What do you mean by "in air"?

Paragraph 2.6 and 3.1: Why did you use 80% v/v methanol as solvent? Please, justify. Furthermore, on what basis were the extraction conditions (time, temperature) decided. I suggest a better discussion of the effect of the solvent used, that is not only linked to the polarity, but other factors such as weakening of the interaction between solute and matrix and swelling of the plant material could also be involved (as discussed in 10.3303/CET1439078).

Paragraphs 3.1: what do you mean with extraction yield? Which is the reference cited in the Table?

Paragraph 3.2 and 4: you cannot compare results expressed in different ways (TEs/g and IC50) In order to do this, I suggest using Trolox equivalents (TEs)/g of extract for both analyses. IC50 depends to the liquid to solid ratio used and so is difficult to be immediately compared by readers. Without the information of the liquid to solid ratio it is impossible to understand the differences.

Discussions: Are the extraction condition used in the cited article the same of your work? If not, you should point out the differences to properly discuss and compare the results.

Which are the characteristics of the extract used in the cited paper? Is it possible to compare with your extract? Please, discuss.

Author Response

All comments have been addressed in the file attached. Thank you.

Reviewer 2 Report

This manuscript was written about the antioxidant activities, total phenolic, total flavonoid, and total alkaloid of selected medicinal plants in Malawi. Please edit the title of the manuscript appropriately because the author didn’t do the experiment to analyze the anticancer activity. It should be antioxidant activity instead of anticancer.

Comments in detail,

1.       What is the positive control for DPPH assay?

2.       What are THPs in Line 102?

3.       Please specify what part of each plant you used in the Plant material.

4.       Please add the results of moisture content.

5.       Please check Fig 1, TPC is in mg QE or GAE?

6.       Fig 2, the letter that indicates significant differences is confusing because we don’t know which one to compare with.

7.       Please add error bars in Fig 1 and Fig 2.

8.       The scientific name of plants in Fig 1 and 2 should be italicized.

9.       Please check the reference format.

10.   Please make the discussion more concise by summarizing your major findings.

Author Response

(The authors gave the same response as above.)

Round 2

Reviewer 1 Report

I thank the authors for their answers

Author Response

All comments are in the documents uploaded.

Reviewer 2 Report

Thank you for revised manuscript about the antioxidant activities, total phenolic, total flavonoid, and total alkaloid of selected medicinal plants in Malawi.

Please check the scientific names of the plants in the abstract. It should be in the same pattern (with or without author name). The abstract should be revised as a summary of your all-key findings. Line 17, change from crude extract to 80% Methanol extract would be preferable. The future work should be put in the discussion instead of abstract. Line 24-25, what is the value of TPC, TFC and TAC that represent the high phytochemical level? Do all these plants exhibit high levels of phytochemicals? Please check the FRAP values again, it seems too high, and the standard deviation is large. Please explain why you need to report again since it appears that almost all the plants in this study have already been reported for their antioxidant activity. It is difficult to follow or understand in the discussion part such as the paragraph in line 300-315. Why does the reference about % yield in the discussion section differ from the reference cited in Table 1? Please check the data in Table 2.

Author Response

All comments  are found in the uploaded document.

Round 3

Reviewer 2 Report

Thank you for revised manuscript about the antioxidant activities, total phenolic, total flavonoid, and total alkaloid of selected medicinal plants in Malawi.

Comments in detail.

1.       Please check if the same scientific name is used more than once, the first letter of the genus may be used as an abbreviation (Line 27).

2.       There are words in the keywords that are duplicated.

3.       Please check the scientific names format (Line 345).

4.       The family name should be capitalized but not italicized (page 2).

5.       Please rewrite the sentence in Line 264-268 and 313-326. If you write 49% methanolic yield, it may be confusing that you are using 49% methanol for extraction. Please check the percentage symbol, it should be no space between it and the number.

6.       Please check FRAP value in Table 2, it may be mg TEAC/ g DW.

7.       Line 356, please check the spelling of further.
